# Comparative Antennal Morphometry and Sensilla Organization in the Reproductive and Non-Reproductive Castes of the Formosan Subterranean Termite

**DOI:** 10.3390/insects12070576

**Published:** 2021-06-24

**Authors:** Paula Castillo, Nathan Le, Qian Sun

**Affiliations:** Department of Entomology, Louisiana State University Agricultural Center, Baton Rouge, LA 70803, USA; pcastillo@agcenter.lsu.edu (P.C.); nle27@lsu.edu (N.L.)

**Keywords:** antennae, sensilla, chemical communication, division of labor, termites

## Abstract

**Simple Summary:**

Insects use antennae to perceive the chemical environment, and olfaction (the sense of smell) is essential for a variety of behavioral responses. Termites are social insects that display a division of labor based on an elaborate caste system consisting of reproductive (queen and king) and non-reproductive individuals (workers and soldiers). Whether these castes have different senses of smell is poorly understood. In this study, we characterized the morphology of antennae in alates (winged reproductives), workers, and soldiers in the Formosan subterranean termite, *Coptotermes formosanus*, and further analyzed the diversity and abundance of the antennal sensilla (sensory receptors) in each caste. We found that both female and male alates had longer antennae and greater numbers of sensilla than workers and soldiers, but all castes possessed the same nine types of antennal sensilla. Each type of sensilla had a specific spatial distribution along the antenna. The quantitative composition of sensilla differed between the reproductive and non-reproductive castes, but few differences were found between female and male alates or between worker and soldier castes. These results suggest that the olfactory morphology is associated with the reproductive division of labor in subterranean termites.

**Abstract:**

Antennae are the primary sensory organs in insects, where a variety of sensilla are distributed for the perception of the chemical environment. In eusocial insects, colony function is maintained by a division of labor between reproductive and non-reproductive castes, and chemosensation is essential for regulating their specialized social activities. Several social species in Hymenoptera display caste-specific characteristics in antennal morphology and diversity of sensilla, reflecting their differential tasks. In termites, however, little is known about how the division of labor is associated with chemosensory morphology among castes. Using light and scanning electron microscopy, we performed antennal morphometry and characterized the organization of sensilla in reproductive (female and male alates) and non-reproductive (worker and soldier) castes in the Formosan subterranean termite, *Coptotermes formosanus* Shiraki. Here, we show that the antennal sensilla in alates are twice as abundant as in workers and soldiers, along with the greater number of antennal segments and antennal length in alates. However, all castes exhibit the same types of antennal sensilla, including basiconicum, campaniformium, capitulum, chaeticum I, chaeticum II, chaeticum III, marginal, trichodeum I, and trichodeum I. The quantitative composition of sensilla diverges between reproductive and non-reproductive castes, but not between female and male alates or between worker and soldier castes. The sensilla display spatial-specific distribution, with basiconicum exclusively and capitulum predominantly found on the ventral side of antennae. In addition, the abundance of chemosensilla increases toward the distal end of antennae in each caste. This research provides morphological signatures of chemosensation and their implications for the division of labor, and suggests future neurophysiological and molecular studies to address the mechanisms of chemical communication in termites.

## 1. Introduction

Antennae are essential peripheral sensory structures in insects. The antennae carry multiple cuticular structures called sensilla, which vary in morphology and harbor different types of neurons for perceiving volatile and non-volatile chemicals, humidity, temperature, and tactile stimuli [1,2]. The majority of insects rely on chemicals for the regulation of fundamental behaviors such as foraging and mating [3]. The morphology of the chemosensory system evolved with the chemical ecology of insect species, resulting in enormously diverse shapes and sizes of antennae and different compositions of antennal sensilla [2,3].

In eusocial insects, chemosensation is crucial for communication among colony members and the modulation of social activities. Eusocial colonies consist of individuals that belong to different castes, with a limited number of individuals dominating reproduction and workers (and soldiers in some species) acting as helpers [4]. The antennal morphology and organization of sensilla often exhibit sex- and caste-specificity, reflecting differential sensory capabilities underlying the division of labor [5,6,7,8,9,10,11]. Sexual dimorphism is common in eusocial Hymenoptera (ants, wasps, and bees), where females are diploid, but males are haploid individuals that develop from unfertilized eggs. In ants, male antennae typically have a shorter scape (i.e., the proximal antennal segment) compared with the female [12]. In addition, male ants in a few species lack basiconicum sensilla, a type of sensilla for perception of cuticular hydrocarbons used for nestmate recognition [5,6,13,14,15]. In the honey bee, *Apis mellifera* Linnaeus, males possess a greater number of sensilla placodea than workers, presumably enhancing their olfactory sensitivity for mate seeking [10]. Differences in antennal morphology and organization sensilla are also associated with the division of labor between non-reproductive castes in several species [8,11]. For example, the guards of a stingless bee, *Tetragonisca angustula* Latreille, have larger antennal surface areas and greater numbers of sensilla, which allow them to detect heterospecific intruders more effectively than other soldiers [11]. Similarly, in the weaver ant, *Oecophylla smaragdina* (Fabricius), major workers that leave the nest and forage possess increased numbers of antennal sensilla than minor workers that remain inside or close to the nest [8].

Compared with eusocial Hymenoptera, termites (Blattodea) are among the few hemimetabolous eusocial insects with diplodiploid sex determination [4]. Unlike hymenopteran colonies that are dominated by females, termite colonies are composed both female and male individuals in each caste. In subterranean termites (Rhinotermitidae), colonies are usually founded by a pair of dispersed alates after a nuptial flight. The alates shed their wings (i.e., dealation), and pair up by tandem running, with a male following a female that releases a sex pheromone; the pair then cooperatively construct a nesting chamber, where they mate and rear the first cohort of brood [16,17,18]. Developing workers then take over the task of brood care, construct foraging tunnels to search for food, and perform hygienic activities. Soldiers companion workers while they forage, protect the colony through aggressive behavior toward intruders, and provide a social buffering effect by alleviating competitor- or predator-induced stress [19,20]. While alates develop compound eyes and ocelli, worker and soldiers are blind [21]. Colony members live in dark underground nests and heavily rely on chemical cues and vibrational communication to organize social behavior [22]. Most previous research on termite chemical communication aimed to characterize the semiochemicals, leading to identification of a number of active compounds, such as the trail pheromones for nestmate recruitment [23], reproductive pheromones that regulate caste differentiation and royal recognition [24,25], sex pheromones for courtship behavior [26,27], and death cues that mediate undertaking behavior [28]. While vibrational behavior is performed in various social contexts such as policing behavior [29], colony defense [20], and reproductive recognition [25], this type of substrate-borne signal is likely perceived by the subgenual organ in the legs [30]. The antennae of termites have been demonstrated to play important roles in chemical-mediated nestmate recognition [31], mating behavior [32], and pathogen detection [33]. However, little is known about the neurophysiological process of chemosensation, mainly due to the lack of morphologically characterized chemosensory system in most termite species.

Termites have moniliform antennae, and each antenna is divided into three basic regions including the scape, the pedicel, and the flagellum composed of varying numbers of flagellomeres (Figure 1A). In termites, antennal segment proliferation occurs at the proximal end of the flagellum upon molting, resulting in an increase in the number of flagellomeres as individuals develop [21]. In the Formosan subterranean termite, *Coptotermes formosanus* Shiraki, the number of antennal segments (including scape and pedicel) varies from 9 in the first instar larvae to 21 in alates in a mature colony [34]. The caste differentiation in termites is primarily a result of developmental plasticity of individuals in response to social and external environments [4], but it remains unknown how organization of antennal sensilla changes as individuals develop into different castes. To date, a few morphological studies on termite antennal sensilla have been conducted [35,36,37], but full characterization of antennal sensilla has been limited to the non-reproductive individuals of *C. formosanus* [38,39,40,41]. Tarumingkeng et al. [38] examined the antennal sensilla of workers in *C. formosanus*, and classified them into chemo- and mechanoreceptors based on the presence or absence of pores in the sensilla. Yanagawa et al. [40] further provided more detailed morphological descriptions of all sensilla in worker antennae, as well as their predicted functions and distribution in each antennal segment. Two studies compared the antennal sensilla between worker and soldier castes, revealing no caste dimorphism in their morphology, distribution and abundance [39,41]. However, no information was available for the antennal morphology and organization of sensilla in the reproductive caste, and the sensory bases of differential behavioral responses among castes were also poorly understood in termites.

The reproductive caste is an integral part for understanding chemosensation and its relationship with the division of labor in termite societies, and a comparison of chemosensory morphology between reproductive and non-reproductive individuals warrants further investigation. In this study, we hypothesized that the worker, soldier, and reproductive castes of termites display differences in antennal morphology and organization of antennal sensilla, which may contribute to their differential sensory capabilities. This hypothesis was tested in *C. formosanus*, one of the most invasive pest species in the world [42]. Specifically, we examined the morphological characteristics of antennae using light microscopy, investigated the composition and spatial distribution of antennal sensilla by scanning electron microscopy, and comparatively analyzed the data in workers, soldiers, female and male alates.

## 2. Materials and Methods

### 2.1. Insects

Female and male alates of *C. formosanus* were collected using ultraviolet light traps (BioQuip, Rancho Dominguez, CA, USA) during the swarm season (May and June) from three populations in Baton Rouge, Louisiana (population A: 30°18′36″ N, 91°05′15″ W; population B: 30°22′14″ N, 91°06′39″ W; population C: 30°21′26″ N, 91°05’27″ W). Upon collection, the alates were processed immediately for analyses of antennal morphology and sensilla. Workers and soldiers were collected from three colonies in New Orleans, Louisiana (colony D: 30°01′26.7″ N, 90°01′01.3″ W; colony E: 29°59′41.3″ N, 90°05′18.4″ W; colony F: 29°54′32″ N, 90°00′32″ W). These colonies were kept at 25 ± 1 °C in clear acrylic containers (38.48 × 45.72 × 22.86 cm), provided with an approximately 4.0 cm layer of organic soil at the bottom and moistened pine wood logs as the food source. The relatively humidity (RH) in each container was monitored weekly, and water was added to maintain 80–99% RH.

### 2.2. Morphometric Analyses

Workers and soldiers with both intact antennae were selected at random from three freshly collected field colonies (*n* = 30 with 10 samples per colony). A total of 29 female and 25 male alates with both intact antennae were selected at random from three populations (female: 9, 10, and 10 individuals from population A, B and C, respectively; male: 5, 10, and 10 individuals from population A, B and C, respectively). To prepare the antennae samples, the termites were anesthetized on ice for at least 5 min and placed in a Petri dish (10 cm in diameter) under a Nikon SMZ1270 microscope coupled with a Nikon DS-Ri2 camera (Nikon instruments Inc., Melville, NY, USA). The antennae were carefully removed from the head of each termite using fine forceps (Dumont #5SF, Fine Science Tools, Foster City, CA, USA) and placed in the Petri dish. Pictures of both antennae (left and right) were taken immediately after dissection with the NIS-Elements software version 4.51 (Nikon instruments Inc., Melville, NY, USA) for further morphometric measurements using ImageJ version 1.52p (National Institutes of Health, Bethesda, MD, USA). The measurements included the total number of flagellomeres, total antennal length, lengths of scape and pedicel, widths of scape and pedicel, and widths of proximal, central, and distal flagellomere. The body lengths were also measured, and the average body length of each caste or sex from pooled samples (*n* = 20 for female and male alates, and *n* = 30 for workers and soldiers) was used to normalize the antennal length of individuals from each caste or sex group.

### 2.3. Antennal Sensilla Analyses

Individuals of each caste with intact antennae were selected for analysis of sensilla using scanning electron microscopy (SEM). Each individual was used for imaging either the ventral or dorsal side of an antenna. A total of 18 female and 18 male alates were analyzed, with *n* = 9 per antennal side each sex (6 and 3 samples from population B and C, respectively). A total of 23 soldiers (*n* = 11 for dorsal side: 5, 3, and 3 samples from colony D, E and F, respectively; *n* = 12 for ventral side: 6, 3 and 3 samples from colony D, E and F, respectively) and 24 workers (*n* = 12 with 6, 3, and 3 samples from colony D, E and F, respectively, for both dorsal and ventral sides) were analyzed. Individuals were anesthetized on ice for at least 5 min. Then, the heads were dissected using fine forceps (Dumont #5SF, Fine Sciences Tools, Foster City, CA, USA), and fixed for 48 h at 4 °C in a mixture of fixatives that contains 4% paraformaldehyde, and 2.5% glutaraldehyde in 0.1 M sodium cacodylate. After fixation, the samples were dehydrated using a gradient of acetone (0, 10%, 25%, 50%, 75%, 100% acetone: water, *v*/*v*) to finally air dry, and then mounted onto an aluminum stud. The samples were metal-coated with platinum-palladium in a EMS550X Sputter coater (Electron Microscopy Sciences, Hatfield, PA, USA), and each sample was mounted either for dorsal or ventral view. Pictures were taken in a JSM-6610LV SEM (JEOL Ltd., Peabody, MA, USA) and utilized for the identification and quantification of antennal sensilla. Types of sensilla were classified based on morphology, and putative functions were further determined based on their location and previously published descriptions [40,41].

### 2.4. Statistical Analyses

All statistical analyses were carried out using the R software version 3.6.3 (The R Foundation, Vienna, Austria) [43], with the exception of principal component analysis (PCA), which was performed using JMP Pro 15 (SAS Institute, Cary, NC, USA) [44]. To check the homogeneity of variances and normality of data distributions, Levene’s tests and Shapiro–Wilk tests were performed, respectively. Generalized linear mixed models (GLMMs) and least square means were utilized to analyze the morphometric data with collection location (i.e., population or colony) coded as a random factor, and caste coded as the fixed factor. Because GLMMs did not fit the antennal sensilla data, the data were pooled from multiple collection locations and analyzed using different methods. Specifically, Kruskal–Wallis and Wilcoxon rank sum tests were performed to evaluate the percentages of sensilla per functional category and the dorsal-ventral distribution of antennal sensilla. In order to compare the relative abundance of each type of sensilla among castes, the proportion of each sensillar type to total sensilla was calculated for each sample. The data were then analyzed by one-way analysis of variance (ANOVA) followed by Tukey’s honestly significant difference (HSD) test. An alpha level of 0.05 was chosen for all tests performed. The relative abundances of nine types of sensilla were also used for PCA to examine the overall patterns across castes.

## 3. Results

### 3.1. Morphometric Analyses

A gross comparison of the antennae from a soldier, worker, and female alate is shown in Figure 1A. Female and male alates did not show difference in their antennal morphology. The antennal cuticle of alates and soldiers showed heavier sclerotization than workers (Figure 1A). Female and male alates possessed significantly more flagellomeres than workers and soldiers (Figure 1B), with the number of flagellomeres per antenna ranging from 17 to 19 in female and male alates, 12 to 13 in soldiers, and 11 to 13 in workers. The most frequent numbers of flagellomeres were 18 in both female and male alates (62.06% and 60.0%, with *n* = 29 and *n* = 25, respectively); 13 in soldiers (66.66%, *n* = 30), and 12 in workers (70.0%, *n* = 30). Female and male alates had significantly longer antennae than workers and soldiers, and soldiers had longer antennae than workers; however, soldiers had the longest antennae in relation to body length among all castes (Table 1). Soldiers and alates did not significantly differ in lengths of scape and pedicel, which in workers were significantly shorter (Table 1). Alates of both sexes had significantly wider scape, pedicel, proximal and central flagellomeres, compared with soldiers and workers; the width of the distal flagellomere, however, did not differ significantly across castes in the measurements performed on left antennae (Table 1). In addition, soldiers had significantly narrower pedicel, proximal and central flagellomeres, compared with other castes (Table 1).

No significant differences were found between the left and right antennae in most of the measurements performed in each caste, but a few exceptions were detected (Table 1). These asymmetries included significantly longer left scape in workers (paired *t*-test; *t =* 2.703, *df =* 29, *p =* 0.0114), wider right central flagellomere in workers (paired *t*-test; *t =* −2.4393, *df =* 29, *p =* 0.0211), wider right scape in female alates (paired *t*-test; *t =* −2.9006, *df =* 28, *p =* 0.0072), and wider right central flagellomere in male alates (paired *t*-test; *t =* −2.6622, *df =* 24, *p =* 0.0136).

### 3.2. Types of Antennal Sensilla

Nine types of sensilla were identified in all castes, including sensilla basiconicum, campaniformium, capitulum, chaeticum I, chaeticum II, chaeticum III, marginal, trichodeum I, and trichodeum II (Figure 2A). These sensilla showed different morphological characteristics, presumably associated with different functions (Table 2). Among them, sensilla basiconicum, chaeticum II, trichodeum I, and trichodeum II have a putative chemosensory function; while chaeticum I, chaeticum III, campaniformium, and marginal have a putative mechanosensory function. Sensillum capitulum is the only type of sensilla found with a putative hygro/thermoreceptive function.

Chemosensilla were the most abundant category of sensilla that constituted approximately 90% of total sensilla in each caste (Figure 2B; 90.10 ± 0.47% in soldiers, *n* = 23; 89.93 ± 0.37% in workers, *n* = 24; 92.16 ± 0.32% in female alates, *n* = 18; 92.09 ± 0.22% in male alates, *n* = 18; dorsal and ventral sides pooled). Alates had significantly higher percentages of chemosensilla than workers and soldiers (soldier-female alate *p =* 0.0014; worker-female alate *p =* 0.0002; soldier-male alate *p =* 0.0009; worker-male alate *p* < 0.0001; Kruskal–Wallis followed by pairwise Wilcoxon rank sum tests with Bonferroni correction) (Figure 2B). The putative mechanosensory sensilla were less abundant and constituted less than 10% of total sensilla in each caste (Figure 2B; 9.56 ± 0.40% in soldiers, *n* = 23; 9.88 ± 0.40% in workers, *n* = 24; 7.77 ± 0.33% in female alates, *n* = 18; 7.87 ± 0.22% in male alates; *n* = 18; dorsal and ventral sides pooled). Compared with alates, significantly higher percentages of mechanosensilla were found in non-reproductives (soldier-female alate *p =* 0.0043; worker-female alate *p =* 0.0013; soldier-male alate *p =* 0.0043; worker-male alate *p =* 0.0013; Kruskal-Wallis followed by pairwise Wilcoxon rank sum tests with Bonferroni correction) (Figure 2B). Sensilla capitulum with a putative hygro/thermoreceptive function did not exceed 0.17% in any caste (Figure 2B). The proportion of this sensillum was significantly reduced in male alates compared with workers and soldiers (*p =* 0.034 in both comparisons; Kruskal–Wallis followed by pairwise Wilcoxon rank sum tests with Bonferroni correction).

### 3.3. Spatial Organization of Antennal Sensilla

The nine types of sensilla showed different patterns of distribution and abundance between the dorsal and ventral side of antennae. In soldiers, female, and male alates, the total number of antennal sensilla did not differ significantly between the dorsal and ventral side of their antennae (Figure 3A; *p* > 0.05; Kruskal–Wallis followed by pairwise Wilcoxon rank sum tests with Bonferroni correction). However, workers had significantly more sensilla on the ventral than dorsal side (Figure 3A; *p =* 0.0069 by Kruskal–Wallis followed by pairwise Wilcoxon rank sum tests with Bonferroni correction). Interestingly, basiconicum sensilla were present exclusively on the ventral side of all castes (Figure 3B), and capitulum sensilla were distributed predominantly on the ventral side in workers, soldiers, and female alates (Figure 3C). A few other types of sensilla also exhibited significant differences between dorsal and ventral distribution in certain castes (Appendix A). For example, chaeticum II sensillum was significantly more abundant on the ventral side of soldiers and workers (for both castes: W = 9, *p =* 0.0003; Wilcoxon rank sum test), while its abundance did not significantly differ between the two sides in alates (*p* > 0.05; Wilcoxon rank sum test) (Appendix A). On the contrary, chaeticum III sensillum was significantly more abundant on the dorsal side in female (W = 63.5, *p =* 0.0411; Wilcoxon rank sum test) and male alates (W = 67, *p =* 0.0190; Wilcoxon rank sum test), but no significant dorsal-ventral asymmetry was found in soldiers and workers (*p* > 0.05; Wilcoxon rank sum test) (Appendix A). In addition, marginal sensilla were more abundant on the dorsal than ventral antennae of soldiers (W = 123.5, *p =* 0.0025; Wilcoxon rank sum test) (Appendix A).

The abundance of total sensilla increased towards the distal end of antennae in all castes (Figure 4). However, not all types of sensilla were ubiquitously present along the antennae. In all castes, sensilla basiconicum, chaeticum I, chaeticum II, trichodeum I, and trichodeum II were distributed in all flagellomeres with increased abundance towards the distal end (Figure 5A–E). However, sensillum capitulum was mostly distributed between the scape and the eighth flagellomere, with decreased abundance towards the distal end (Figure 5F). Sensilla chaeticum III and campaniformium were predominantly found on scape and pedicel (Figure 5G,H), while marginal sensilla were mostly present on the scape and pedicel of all castes, with a distribution between flagellomeres 4 to 7 in workers (Figure 5I).

Trichodeum I was the most abundant type of sensilla found in all castes, corresponding to 49.55 ± 0.65%, 49.44 ± 1.12%, 46.83 ± 0.36% and 46.63 ± 0.42% (mean ± SE) of the total antennal sensilla in soldiers, workers, female and male alates, respectively, on the ventral side (Figure 6B). The second most abundant type of sensilla was chaeticum II, which ranged from 34.22 ± 1.17% in workers up to 39.76 ± 0.40% in female alates (Figure 6B). Sensillum chaeticum I and trichodeum II were the third and fourth most abundant types of sensilla, respectively. Sensillum chaeticum I ranged from 6.76 ± 0.26% in female alates up to 8.20 ± 0.34% in workers; sensillum trichodeum II ranged from 4.00 ± 0.20% in male alates up to 5.10 ± 0.28% in workers (Figure 6B). Sensillum basiconicum, capitulum, chaeticum III, campaniformium, and marginal were the least abundant types of sensilla, which together did not exceed 3.0% of the total antennal sensilla (Figure 6B). The rank for the abundance of the nine types of sensilla on the dorsal side followed the same order (Figure 6A).

### 3.4. Composition of Antennal Sensilla in Different Castes

The total number of sensilla was significantly higher in female and male alates than workers and soldiers (Figure 3A). When normalized to total sensilla on the dorsal side, no significant differences across castes were found for sensilla campaniformium, capitulum, chaeticum III, and trichodeum II (Figure 6A). However, workers and soldier had significantly higher proportions of sensilla chaeticum I than alates (soldier-female *p =* 0.031; worker-female *p =* 0.01; soldier-male *p =* 0.015; worker-male *p =* 0.004; one-way ANOVA followed by Tukey’s HSD), while increased percentages of sensilla chaeticum II were found in female and male alates (soldier-female *p =* 0.007; worker-female *p =* 0.0002; soldier-male *p =* 0.005; worker-male *p =* 0.0002; one-way ANOVA followed by Tukey’s HSD) (Figure 6A). In addition, soldiers had a significantly higher proportion of marginal sensilla than female alates (*p =* 0.006; one-way ANOVA followed by Tukey’s HSD), and sensilla trichodeum I were detected in a significantly higher proportion in workers than both female and male alates (worker-female *p =* 0.009; worker-male *p =* 0.009; one-way ANOVA followed by Tukey’s HSD) (Figure 6A). For the ventral sensilla, no significant differences across castes were found for sensilla campaniformium, chaeticum I, II, and III, marginal sensillum, and sensillum trichodeum I (Figure 6B). However, a significantly lower proportion of sensillum basiconicum was found in soldiers compared with male alates (*p =* 0.0111; one-way ANOVA followed by Tukey’s HSD) (Figure 6B). Workers possessed a higher proportion of sensillum trichodeum II than other castes (worker-female alate: *p =* 0.0022; worker-male alate: *p =* 0.0031; worker-soldier: *p =* 0.0019; one-way ANOVA followed by Tukey’s HSD), and sensillum capitulum was found in significantly higher proportions in soldiers and workers than female and male alates (soldier-female alate: *p =* 0.0300; soldier-male alate: *p =* 0.0022; worker-female alate: *p =* 0.0040; worker-male alate: *p =* 0.00023; one-way ANOVA followed by Tukey’s HSD) (Figure 6B). When the proportions of all nine types of sensilla were included, PCA revealed a divergence between the reproductive (female and male alate) and non-reproductive (worker and soldier) castes; however, female and male alates were not distinguished from each other, and there was no separation between worker and soldier castes (Figure 7; the first two principal components explained 58.8% and 50.5% of the total variance for dorsal and ventral sensilla, respectively).

## 4. Discussion

### 4.1. Antennal Morphology

Our results revealed caste-specific differences in antennal morphology, but no major sex-specific differences were found in alates. The varying number of flagellomeres reflects the age and developmental status of these castes, as flagellomere number increases with the development of termites [21,34]. In subterranean termites, soldiers differentiate from workers (and additionally the second instar larvae in incipient colonies of *C. formosanus*), and both workers and soldiers are immature; alates are the only mature individuals that undergo more molts than most other individuals [4,34]. The caste development and age composition of individuals is also dependent on colony age [34], and the colonies used for non-reproductive samples in our study were likely mature colonies (>5 years), based on their antennal flagellomere ranges (11–13 in workers and 12, 13 in soldiers) and previously published information in *C. formosanus* [34].

It is worth noting that soldiers had longer antennae in relation to body length than workers and alates, and more heavily sclerotized antennae compared with workers, which highlight the defensive role of the soldier caste [19,45]. The longer antennae may facilitate perception of environmental information in a relatively wider range, which allows soldiers to efficiently detect threatening cues from predators and competitors. The elongation in antennae is among several specialized morphological characteristics in *C. formosanus* soldiers, such as enlarged and heavily sclerotized heads and mandibles for physical defense, and the presence of frontal gland for chemical defense [46]. Such a modification in soldier antennal morphology may be widespread in termites, which awaits further investigation across taxonomic groups.

Bilateral (left/right) asymmetry in antennal length was not found in any caste examined in this study; however, asymmetries were detected in a few morphometric measurements, such as the length of scape in workers, the width of scape in female alates, and the width of central flagellomere in workers and male alates (Table 1). The proximate causes and functions are unknown for these bilateral asymmetries. Our observations are possibly a case of fluctuating asymmetry (i.e., small, random deviations from perfect bilateral symmetry), which is a result of gene–environment interaction and an indicator of environmental stress during development [47,48]. The hemimetabolism has predisposed termites to display developmental plasticity in response to social and external environments, which are important for caste development [4]; however, additional empirical evidence is needed to determine the genetic and environmental influences on antennal development in termites.

### 4.2. Antennal Sensillar Types and Spatial Organization

We classified the antennal sensilla into nine morphological types, corroborating previously published descriptions by Yanagawa et al. [40] (but different nomenclatures are used by Deng et al. [39] and Fu et al. [41]). These sensilla belong to three functional groups, and the abundance analysis (Figure 2B) indicates that the antennae of *C. formosanus* are mainly chemosensory organs, but also sensitive to other sensory information such as mechanical stimuli, humidity, and temperature.

A few types of sensilla were differentially distributed on the dorsal and ventral surfaces of antennae. In particular, basiconicum sensilla were located only on the ventral side in each caste. A number of electrophysiological studies have shown that basiconicum sensilla in ants are responsive to cuticular hydrocarbons [14,15,49], an important class of chemicals for communication of species, colony, caste, and reproductive status in social insects [50,51,52]. Sexual dimorphism and spatial-specific distribution of basiconicum sensilla have been reported in ants. These sensilla are absent from males in the red imported fire ant, *Solenopsis invicta* Buren, the Japanese carpenter ant, *Camponotus japonicus* Mayr, and the clonal raider ant, *Ooceraea biroi* (Forel) [5,6,13]. In *S. invicta*, most basiconicum sensilla are clustered on the ventral and medial surface of the distal region of antennal club [13]; in *C. japonicus*, the abundance of basiconicum sensilla increases towards the distal antennal segments [6]; and in *O. biroi*, these sensilla are restricted to the ventral side of the antennal club [5]. Such a sex- and spatial-specific distribution corresponds to the limited social communication in male ants, as well as the social behavior by workers that use restricted antennal club regions to contact nestmates and brood. Termite colonies are composed of female and male individuals both extensively participating in social interactions. In *C. formosanus*, the ventral-biased distribution in each caste is consistent with our observation that individuals often use the ventral surface of antennae to contact the body of their nestmates, suggesting that basiconicum sensilla play a role in social communication in this species. However, the odorant receptors housed in and the function of basiconicum sensilla are yet to be determined in termites.

The capitulum is a putative hygro/thermo sensillum with a ventral-biased distribution in workers, soldiers, and female alates. In subterranean termites, environmental temperature, humidity, and substrate moisture are important factors influencing survival and foraging behavior of workers and soldiers [53,54], and electrophysiological evidence shows that *C. formosanus* worker antennae are responsive to humidity changes [55]. The ventral-biased distribution of capitulum sensilla suggests termites may use them for moisture and/or temperature detection by contacting the substrate, but it is puzzling that the distribution pattern was different in male alates.

In all *C. formosanus* castes examined in this study, antennal sensilla were unevenly distributed across the flagellum with an increase in the abundance of chemosensilla towards the distal end (Figure 5), indicating the distal flagellomeres are more sensitive to environmental odorants. Although intact antennae were analyzed in this study, we have observed that field-collected workers and soldiers often had partially damaged antennae, presumably due to aggressive interactions with other termite colonies or predators. Although the abundance varies along flagellum, all four putative chemosensilla (i.e., basiconicum, chaeticum II, trichodeum I, and trichodeum II) were found in nearly every flagellomere (Figure 5). This suggests that partial damage to antennae may lead to impaired sensitivity but not functional loss of chemosensation. An interesting and widely documented behavior is the antennal cropping by both female and male dealates (i.e., wingless primary reproductives) after colony foundation [56,57,58]. In the congeneric species *C. gestroi* and *C. lacteus*, an average of 7.3 and 5.1 flagellomeres, respectively, has been documented to be cropped in dealates [56]. Such a behavior was proposed to reduce pheromonal sensitivity upon the transition from nuptial flight in the open environment to reproduction in the enclosed nest [56], and this “sensitivity hypothesis” is supported by the results of our sensilla abundance analysis. Additional research on pheromone perception in the antennal sensilla and pheromone processing in the brain is required to provide insights into the neural mechanisms underlying the behavioral transition in reproductives.

### 4.3. Comparison of Antennal Sensilla among Castes

Overall, we found no difference in the diversity of antennal sensilla among the worker, soldier, and reproductive castes in *C. formosanus*; in addition, our results show quantitative differences in sensilla between the non-reproductive and reproductive castes, but not between female and male alates (Figure 3, Figure 6 and Figure 7). The discrepancy in the quantitative composition of antennal sensilla reflects differential sensitivity to environmental stimuli, which may underlie the division of labor among castes in *C. formosanus* colonies.

In subterranean termites, alates are the only individuals that are exposed to the open environment, and these individuals have an expanded behavioral repertoire compared with other castes [16,17,18]. The greater antennal sensilla abundance in alates may allow them to perceive a greater range of information. Before colony foundation, these individuals must detect suitable environmental conditions for dispersal [59], sex pheromones for mate seeking [27,32], and available food and moisture for nest construction [16]. During the incipient stage of a colony, the primary reproductives are expected to recognize chemical and/or tactile signals from the offspring for brood care [60], and cuticular hydrocarbons for nestmate recognition [61]. By contrast, workers and soldiers live in enclosed underground nests and perform a suite of collective behavior related to foraging, colony hygiene, and defense [4,19]. The smaller number of antennal sensilla in the non-reproductive individuals, as a result of shorter antennae due to developmental immaturity, may reflect an individual-level reduction in sensitivity to environmental cues related to dispersal and mating.

When the proportions of each sensillar type to total sensilla were compared among castes, alates displayed an enriched dorsal distribution of chaeticum II (Figure 6A), a putative chemosensillum most abundant on the distal flagellomere (Figure 5). This finding, together with the commonly observed antennal cropping behavior [56], suggest that chaeticum II may be involved in mating seeking and nesting site searching in the dispersal reproductives. In termites, the female displays calling behavior by releasing a sex pheromone, and leads the search for a suitable nesting location, while the male follows in tandem upon detection of the female [17,27]. Therefore, sexual dimorphism in chemosensation is expected in alates. However, in our study, no sex-specificity was found in the organization of antennal sensilla. The chemoreceptors and other chemosensory organs such as mouthparts should be examined for a better understanding of the sex-specific behavior.

Compared with the alates, chaeticum I was enriched on the dorsal side and capitulum was enriched on the ventral side of workers and soldiers (Figure 6). Chaeticum I sensillum has a predicted mechanosensory function and capitulum is likely responsive to humidity/moisture and thermal changes [40]. Such an organization in the non-reproductive individuals highlights their sensitivity to mechanical stimuli, which include tactile information for nestmate interactions and physical disturbance to the nest caused by biotic and abiotic factors [62,63], as well as environmental humidity and moisture, which are crucial for the survival of subterranean termites [53,54]. Despite many behavioral differences between workers and soldiers, these two castes share similar quantitative composition of antennal sensilla (Figure 6 and Figure 7), which is consistent with previously reported findings by Fu et al. [41]. Compared with soldiers, workers only exhibited a greater proportion of trichodeum II sensilla on the ventral but not dorsal side. Trichodeum II is a putative chemosensillum, but its biological function is yet to be determined. The results imply that, other than structural bases, molecular and neural level differences in chemosensation may contribute to the division of labor among *C. formosanus* castes.

## 5. Future Directions

Our results provide an increased understanding of the morphological basis of chemosensation and its relationship with the division of labor in subterranean termites. We suggest additional studies across termite taxa for comparative analysis of chemosensory morphology. Although *C. formosanus* castes exhibit several differences in the morphology of antennae and the abundance of sensilla, the structural signatures are among many other chemosensory characteristics that may show more prominent caste- and/or sex-specificity. To elucidate the chemosensory bases of the division of labor in termites, future investigations are required to address the neurophysiological and molecular mechanisms governing chemical perception in the peripheral sensory system and information processing in the brain. We propose immediate next steps that include functional tests of various types of sensilla using single sensillum recording (SSR), and transcriptomic analysis of chemosensory gene repertoires in multiple castes. In addition, morphological and molecular characterization of other sensory organs, particularly the mouthparts, may offer further information on chemical communication in termites. Compared with current knowledge in social Hymenoptera [64,65], electrophysiological studies and functional genetics in termites are still in their infancy. However, recent utilization of neurophysiological approaches such as SSR [66] and immunohistochemistry [67], the development of next-generation sequencing tools [68,69], and the successful application of the RNA interference technique [70,71], offer important avenues for the future exploration of chemosensation and division of labor in termites.

## Figures and Tables

**Figure 1 insects-12-00576-f001:**
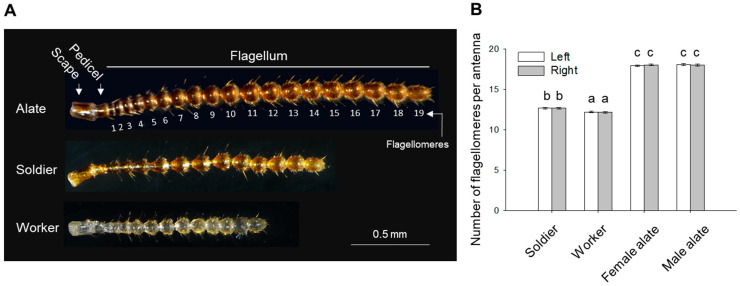
Overall antennal morphology in *C. formosanus*. (**A**) Representative images of antennae from a female alate, a soldier, and a worker. The antenna is divided into three main regions: scape, pedicel, and flagellum, which is composed of flagellomeres numbered from the proximal to distal end, as exemplified in the alate. (**B**) Numbers of flagellomeres on the left and right antennae in soldiers, workers, female and male alates. Bars represent mean ± standard error (SE). Groups denoted with the same letter are not significantly different (generalized linear mixed model (GLMM), *p* > 0.05; *n* = 30 for soldiers and workers, *n* = 29 for female alates, *n* = 25 for male alates).

**Figure 2 insects-12-00576-f002:**
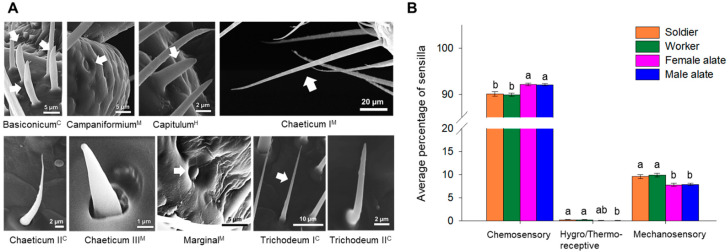
Morphological types of sensilla and functional categories. (**A**) Nine types of sensilla identified in all castes C putative chemosensory sensilla; H/T: putative hygro/thermoreceptive sensilla; M: putative mechanosensory sensilla). (**B**) Proportions of sensilla per functional category. Bars represent percentage of the different types of sensilla combined per functional category (mean ± SE; *n* = 24 for workers, *n* = 23 for soldiers; and *n* = 18 for both female and male alates). For each functional category, groups denoted with the same letter are not significantly different (*p* > 0.05, Kruskal–Wallis followed by pairwise Wilcoxon rank sum tests with Bonferroni correction).

**Figure 3 insects-12-00576-f003:**
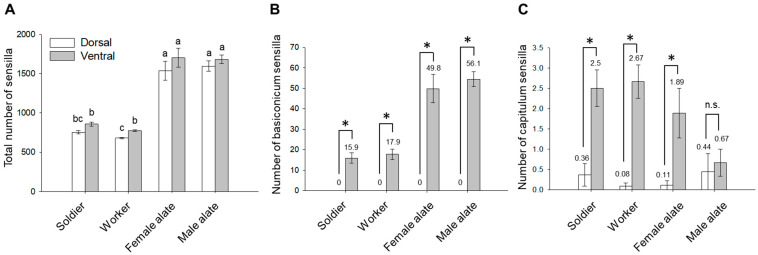
Dorsal and ventral distribution of antennal sensilla in *C. formosanus* castes. (**A**) Total number of sensilla on dorsal and ventral sides per antenna (mean ± SE). Groups denoted with the same letter are not significantly different (*p* > 0.05, Kruskal–Wallis followed by pairwise Wilcoxon rank sum tests with Bonferroni correction). (**B**,**C**) show number of basiconicum and capitulum sensilla per antenna, respectively, distributed on each side. Means are presented above bars. Bars represent mean ± SE (*, *p* < 0.05; n.s., not significant, *p* > 0.05; Wilcoxon rank sum test). Dorsal side: *n* = 11, 12, 9, and 9 for soldier, worker, female and male alate, respectively; Ventral side: *n* = 12, 12, 9, and 9 for soldier, worker, female and male alate, respectively.

**Figure 4 insects-12-00576-f004:**
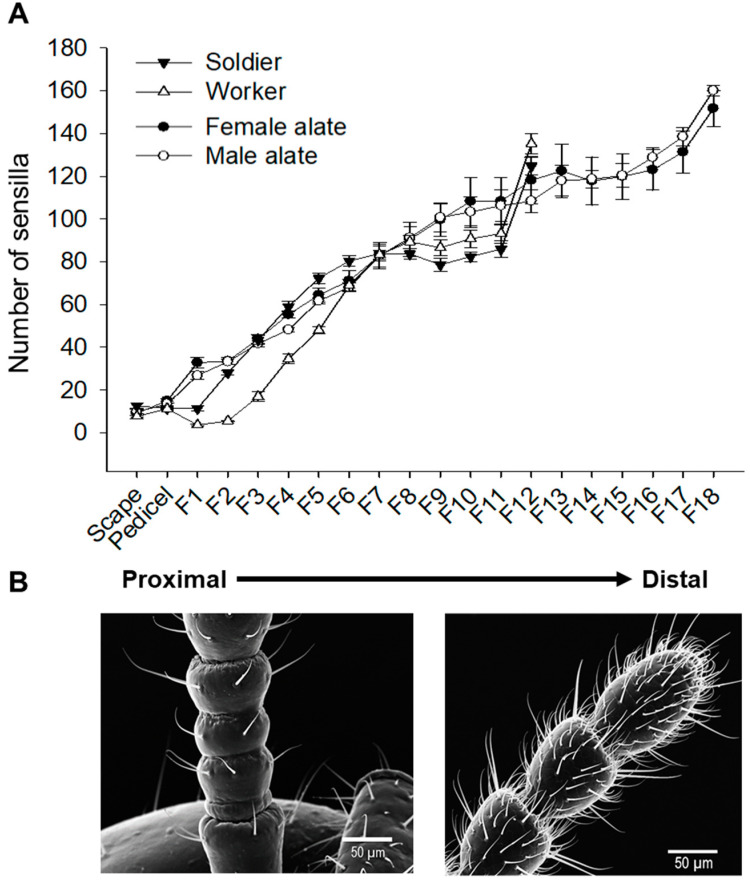
Distribution of total sensilla along the antennae in *C. formosanus* castes. (**A**) Ventral side abundance of sensilla per antennal segment. F1-F18 denote flagellomeres numbered from proximal to distal end. All alate samples analyzed here had 18 flagellomeres, while worker and soldier samples had 12 flagellomeres. Data are presented as mean ± SE (*n* = 12 for soldiers and workers, *n* = 9 for female and male alates). (**B**) A worker antenna representing the distribution of sensilla at the distal and proximal ends.

**Figure 5 insects-12-00576-f005:**
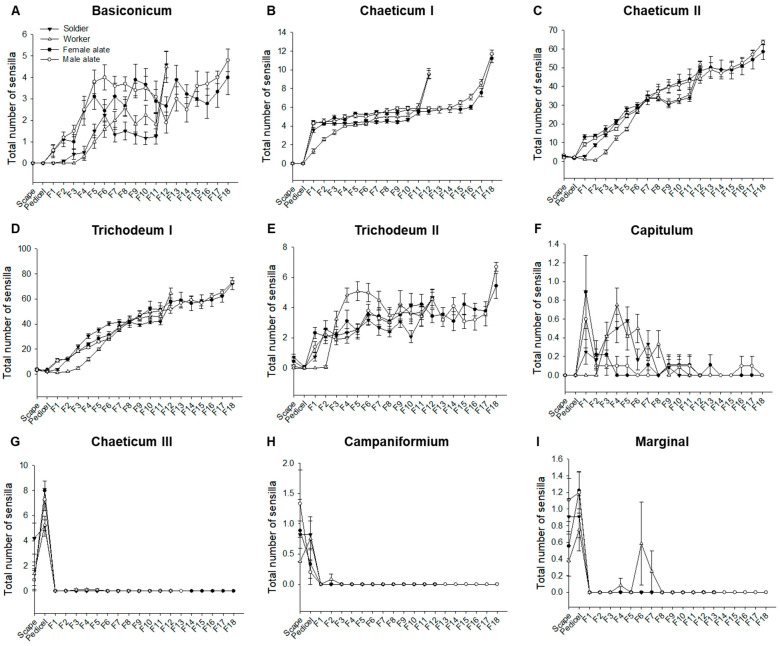
Distribution of nine types of sensilla along the ventral side of the antennae in *C. formosanus* castes. The numbers of sensilla basiconicum (**A**), chaeticum I (**B**), chaetucum II (**C**), trichodeum I (**D**), trichodeum II (**E**), capitulum (**F**), chaeticum III (**G**), campaniformium (**H**), and marginal (**I**) located on the scape, pedicel, and each flagellomere are shown (mean ± SE, *n* = 12 for soldiers and workers, *n* = 9 for female and male alates).

**Figure 6 insects-12-00576-f006:**
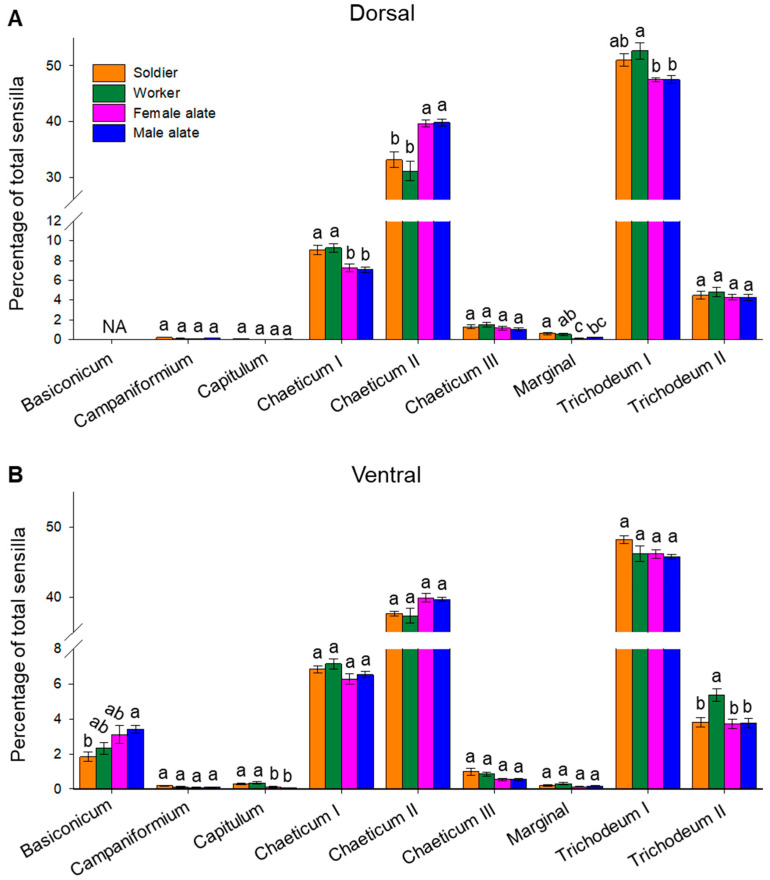
Comparison of relative abundance of nine sensillar types among castes. The plot shows percentage of each type of sensilla to total sensilla on the dorsal (**A**) and ventral (**B**) side. Bars represent mean ± SE. For each type of sensilla, groups denoted with the same letter are not significantly different (*p* > 0.05, one-way analysis of variance (ANOVA) followed by Tukey’s honestly significant difference (HSD); Dorsal side: *n* = 11, 12, 9, and 9 for soldier, worker, female and male alate, respectively; Ventral side: *n* = 12, 12, 9, and 9 for soldier, worker, female and male alate, respectively; NA: not available).

**Figure 7 insects-12-00576-f007:**
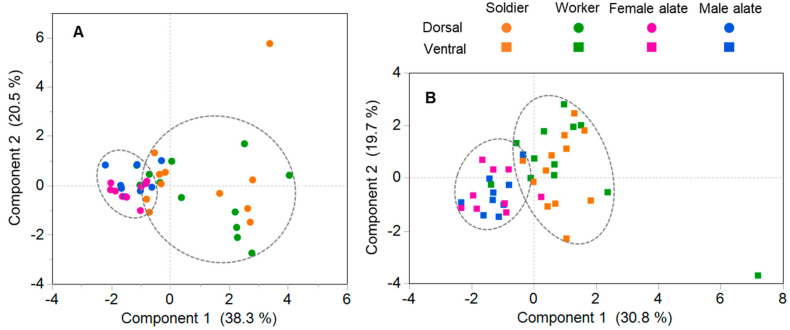
Principal component analysis (PCA) of dorsal (**A**) and ventral (**B**) sensilla in different castes. All nine types of sensilla were included, and percentages of each sensillar type to total sensilla were used in the analysis. The two gray ellipses in each panel encircle approximate 90% samples in the reproductive (female and male alate) and non-reproductive (soldier and worker) castes, respectively. (Dorsal side: *n* = 11, 12, 9, and 9 for soldier, worker, female and male alate, respectively; Ventral side: *n* = 12, 12, 9, and 9 for soldier, worker, female and male alate, respectively).

**Table 1 insects-12-00576-t001:** Morphometric comparison of antennae among *C. formosanus* castes. Measurements (mean ± SE) from left and right antennae were analyzed separately to compare soldier, worker, female and male alate samples. Groups denoted with the same letter in each row are not significantly different (GLMM, *p* > 0.05; *n* = 30 for soldiers and workers, *n* = 29 for female alates, *n* = 25 for male alates).

	Side	Soldier	Worker	Female Alate	Male Alate
Total antennal length (mm)	Left	1.643 ± 0.019 ^(b)^	1.277 ± 0.017 ^(a)^	2.309 ± 0.014 ^(c)^	2.284 ± 0.020 ^(c)^
	Right	1.648 ± 0.019 ^(b)^	1.265 ± 0.020 ^(a)^	2.335 ± 0.016 ^(c)^	2.284 ± 0.026 ^(c)^
Antennal length normalized to body length (%)	Left	45.78 ± 0.540 ^(b)^	36.42 ± 0.555 ^(a)^	34.45 ± 0.215 ^(a)^	35.61 ± 0.310 ^(a)^
	Right	45.93 ± 0.540 ^(b)^	36.11 ± 0.660 ^(a)^	34.84 ± 0.251 ^(a)^	35.60 ± 0.409 ^(a)^
Length of scape (µm)	Left	163.50 ± 3.053 ^(ab)^	159.13 ± 2.750 ^(a)^	173.16 ± 4.160 ^(c)^	179.66 ± 2.360 ^(bc)^
	Right	165.43 ± 2.906 ^(b)^	148.27 ± 3.635 ^(a)^	179.03 ± 2.202 ^(b)^	173.08 ± 2.974 ^(b)^
Length of pedicel (µm)	Left	90.23 ± 1.506 ^(b)^	84.50 ± 0.816 ^(a)^	91.97 ± 0.131 ^(b)^	92.52 ± 1.372 ^(b)^
	Right	89.40 ± 1.252 ^(b)^	82.47 ± 1.462 ^(a)^	95.38 ± 1.285 ^(b)^	94.72 ± 1.285 ^(b)^
Width of scape (µm)	Left	95.60 ± 0.601 ^(a)^	94.90 ± 0.929 ^(a)^	120.38 ± 1.097 ^(b)^	124.00 ± 1.350 ^(c)^
	Right	96.20 ± 0.904 ^(a)^	93.03 ± 1.287 ^(a)^	124.55 ± 1.344 ^(b)^	121.40 ± 1.316 ^(b)^
Width of pedicel (µm)	Left	72.90 ± 0.597 ^(a)^	78.10 ± 0.422 ^(b)^	94.36 ± 0.984 ^(c)^	93.55 ± 0.930 ^(c)^
	Right	72.90 ± 0.524 ^(a)^	77.23 ± 0.644 ^(b)^	95.45 ± 0.628 ^(c)^	93.76 ± 0.722 ^(c)^
Width of proximal flagellomere (µm)	Left	64.03 ± 0.885 ^(a)^	71.20 ± 0.643 ^(b)^	86.80 ± 1.235 ^(c)^	87.41 ± 1.604 ^(c)^
	Right	65.83 ± 1.092 ^(a)^	72.13 ± 0.847 ^(b)^	88.97 ± 1.283 ^(c)^	86.40 ± 1.786 ^(c)^
Width of central flagellomere (µm)	Left	90.83 ± 0.924 ^(a)^	93.90 ± 1.004 ^(a)^	115.45 ± 1.294 ^(c)^	109.40 ± 1.094 ^(b)^
	Right	91.63 ± 0.931 ^(a)^	96.27 ± 0.953 ^(b)^	115.55 ± 1.110 ^(c)^	112.76 ± 1.482 ^(c)^
Width of distal flagellomere (µm)	Left	80.17 ± 0.803 ^(a)^	83.07 ± 0.694 ^(a)^	83.34 ± 1.073 ^(a)^	83.68 ± 1.174 ^(a)^
	Right	80.60 ± 0.592 ^(a)^	83.50 ± 0.805 ^(b)^	82.41 ± 0.873 ^(ab)^	83.48 ± 0.818 ^(ab)^

**Table 2 insects-12-00576-t002:** Characteristics of sensilla. Morphological measurement data (mean ± SE, *n* = 3) were obtained from workers. Putative functions were determined based on morphology and location of sensilla and previously published descriptions [40,41].

Sensillar Type	Morphological Characteristics	Putative Function	Total Length (µm)	Basal Diameter (µm)	Tip Diameter (µm)	Socket Diameter (µm)	Central Diameter (µm)
Basiconicum	Short, blunt-tipped, cylindrical-shaped with a narrow tip	Chemosensory	12.916 ± 0.685	2.129 ± 0.067	0.504 ± 0.068	3.559 ± 0.070	-
Campaniformium	Smooth-surfaced oval-shaped pore	Mechanosensory	4.472 ± 0.127	-	-	-	1.702 ± 0.088
Capitulum	Cone-shaped, wider at the base and narrowed towards the tip	Hygro/thermoreceptive	6.579 ± 0.162	2.483 ± 0.087	0.608 ± 0.032	3.970 ± 0.226	-
Chaeticum I	Longest sensilla; the wide base gives support to its cylindrical shape that ends in a thin tip	Mechanosensory	114.649 ± 7.595	4.209 ± 0.310	0.618 ± 0.007	5.523 ± 0.226	-
Chaeticum II	Wider at the base and narrower towards the tip; slightly curved towards the surface of the antennae	Chemosensory	18.790 ± 2.405	1.986 ± 0.099	0.315 ± 0.006	2.593 ± 0.106	-
Chaeticum III	Short, cone-shaped sensilla located in a socket-like structure	Mechanosensory	5.860 ± 0.500	1.636 ± 0.029	0.397 ± 0.045	2.776 ± 0.150	-
Marginal	Smooth-surfaced dome in a socket	Mechanosensory	-	-	-	3.360 ± 0.697	1.899 ± 0.464
Trichodeum I	Similar in shape to Chaeticum II but straight, without curving towards the surface of the antennae	Chemosensory	49.093 ± 1.069	2.473 ± 0.061	0.455 ± 0.015	3.546 ± 0.302	-
Trichodeum II	Wider than trichodeum I, slightly curved towards the surface of the antennae	Chemosensory	11.520 ± 0.207	1.840 ± 0.162	0.291 ± 0.061	2.487 ± 0.235	-

## Data Availability

The data presented in this study are available in the Appendix A.

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
