# Peer review of "Comparative Antennal Morphometry and Sensilla Organization in the Reproductive and Non-Reproductive Castes of the Formosan Subterranean Termite"

_insects, 2021, doi:10.3390/insects12070576_

Round 1

Reviewer 1 Report

It was a real pleasure to review this paper about the morphometry and the sensilla organization of antennae in termites. The authors are right, this kind of study are rare despite of its primary importance to understand the mechanisms of communication in social insects. A lot of information and descriptions are available in this article. Moreover, it is clearly exposed and well described. My only major comment concern the statistical analyses about the sampling. It is still unclear despite my numerous readings. Indeed, it is said that several populations were used, and it was apparently correctly handed for the GLMMs analyses by adding the colony/populations as random factors. But what about the means for the other analyses. In fact, 3 colonies/populations were collected with an average of 10 individuals per colonies. But then it was pooled together to calculate the means. It would have been more correct to make a mean per colony, and then to make the statistical comparisons with 3 points only (which seems not enough). Studying social insects imply to take care of the colonial level where sampling of one colony represent only one point. Taking several individuals from the same colony should be seen as a replicate of the same sample. So, to avoid pseudo-replication it is important to reduce the dataset to one value per colony, unless it is considered with random factors. So, cautions must be taken about few analyses and authors should at least stipulate it in the manuscript.

Authors do not consider vibrations which is particularly important for communication in subterranean species and specifically for termites. They should consider adding few references in the introduction and to discuss their results on mechanosensory sensilla with this literature. Moreover, it will be interesting to make a small paragraph on the comparison between what is known for Isopterans compared to Hymenopterans. Few comparisons are sometimes drawn but it would be interesting to add the evolutionary prospect into the discussion.

It would be informative to visualize the statistical values into the figure 2B.

In the discussion, two negative forms are in the same sentence: "However, in our study, no sex-specificity was not found in the organization of antennal sensilla.". Please fix it.

Reviewer 2 Report

Please see comments in attached PDF.

Interesting and useful study.

Please check journal required format.  References for this journal are not superscripted and in brackets.  This needs to be changed. [1,2].

Table 1 caption is out of place.  Fix this.

Table 2 is really hard to discern where morphological characteristic descriptions start and end.  Please fix this table.  Probably smaller font.

Please state authority after first mention of genus species throughout text.  Example Coptotermes formosanus Shiraki

Consider more colored figures.  Change some of the black bars to pattern so reader can see SE bars.

Adjust wideness of bars in figure 6.  They are too cramped together.  hard to read statistical lettering.

Format references section cording to journal rules.  
